# Work Conditions of Italian Nurses and Their Related Risk Factors: A Cohort Investigatory Study

**DOI:** 10.3390/diseases10030050

**Published:** 2022-08-03

**Authors:** Elsa Vitale

**Affiliations:** Local Health Company Bari, 70124 Bari, Italy; vitaleelsa@libero.it

**Keywords:** anxiety, body mass index, depression, nurse, sex, shift, stress, work experience

## Abstract

(1) Background: Nursing is a satisfying employment pathway, as nurses preserve lives, but it is also considered one of the most stressful care professions. Nursing is a lifesaving and highly satisfying profession, yet it is considered one of the most stressful occupations. This study aimed to assess differences in anxiety, depression and stress states among nurses according to gender, work history, shift and body mass index (BMI) characteristics. (2) Methods: An online questionnaire was addressed to all Italian nurses during May 2022, investigating anxiety, depression and stress conditions according to the Depression, Anxiety and Stress Scale-21 Items (DASS-21) and sampling characteristics, such as sex, work experience, shift and BMI values. (3) Results: A total of 408 Italian nurses answered the questionnaire. Significant differences between the following were reported: anxiety levels and work experience, since nurses employed over 6 years reported higher anxiety levels than their younger colleagues (*p* = 0.035); depression levels and sex (*p* = 0.017), as females reported higher depression levels than males; and also between depression levels and BMI levels (*p* = 0.003), as 5.90% of overweight and 2.50% of obese participants reported extremely severe depression. By considering stress levels according to sampling characteristics, significant differences were registered according to BMI levels (*p* < 0.001), as overweight subjects reported higher stress levels (7.40%) than the other subgroups. Finally, significant associations were recorded between anxiety, depression and stress conditions with sex, work experience, shift and BMI. (4) Conclusions: The data were in agreement with the current literature, indicating that nurses might take care not only of their patients but also of themselves, in both the physical and mental aspects.

## 1. Introduction

Nursing is a satisfying employment pathway, as it preserving lives. On the other hand, it is also one of the most stressful care professions. Every day, nurses deal with physical and emotional situations, which are often combined with several factors, such as understaffing due to the low availability of funds in the healthcare system, heavy workloads, and the increasing utilization of sophisticated healthcare technologies [1]. Therefore, nursing is considered as the most stress-related profession in the entire healthcare system [2]. Research has highlighted that nurses are stressed and overemployed, with a poor perception of their own safety in their many working environments [3]. Stressed workers are susceptible to harm and poor health, leading to absenteeism and reduced productivity [4]. In stress conditions, corticosteroids regulate learning and memory processes due to a rapid alternating hypothalamic–pituitary–adrenal (HPA) axis activation and inhibition, which depend on the phase of the pulse [5,6] and are also affected by circadian rhythms [7,8].

The literature demonstrates a very strong relationship between nursing, working conditions and psychological conditions, such as stress, anxiety and depression [9,10,11,12,13], which negatively influence biological rhythms [9,10] and increase the risk of a hostile work–life balance [14,15], for example by increasing the BMI [16] and the possibility of work-related injury [17,18,19]. Several studies have investigated the association between employment and psychological conditions, particularly stress, anxiety and depression [20,21,22,23,24,25,26]. In all healthcare organizations, shift work, particularly in nursing employment, was considered indispensable for ensuring continuity in patient care, also including night shifts [27]. Both shift and night shift work were considered as the most common factors in the dysregulation of physiological circadian rhythms, influencing both physical and psychological well-being and reducing work performance [28,29,30]. Some studies recognized a maladaptation syndrome associated with shift work, involving an impaired sleeping/waking pattern, gastrointestinal disorders and an increased risk of cardiovascular diabetic diseases [31,32,33,34,35,36,37,38,39], which also seemed to be associated with increasing variations in BMI values [40,41,42,43].

Depression is a very common mental health disorder in several working environments [44,45], and women seem to be at a major risk of becoming depressed [46,47]. A Chinese cross-sectional study explored the prevalence of depression among 1592 nurses, reporting 25.1% rate of moderate and severe depressive symptoms [48]. This trend is very common all around the world, highlighting the prevalence of depressive symptoms in the female population [49,50,51]. Due to heavy workloads, nigh shifts, and little autonomy in their professions, nurses seem to be more stressed in their working environments, and this is also due to poor working organization, design, management, unsatisfactory working conditions and a lack of support from colleagues and supervisors [52].

The literature has highlighted strong associations between work-related stress and mental health problems and similar symptoms [9,10,11,12,13,14,15,16,17,18,19]. However, very few studies have focused on all these variables simultaneously in order to assess how the nursing profession is related to psychological conditions.

Therefore, the present study aimed to assess any differences in anxiety, depression and stress conditions among nurses according to sex, work experience, shift and BMI characteristics. Additionally, this research also evaluated any associations between sampling characteristics and psychological conditions in order to establish the ultimate existence of a potential risk factor in the nursing profession, which could also contribute to worse mental health conditions. Specifically, the purpose of the present research was to investigate whether the nursing profession could itself pose a risk of developing negative psychological conditions.

## 2. Materials and Methods

### 2.1. Study

An online cohort observational study was conducted during May 2022. The questionnaire was created ad hoc through the Google function of Google Modules and publicized though some nursing socials and pages found on Facebook and Instagram, such as #noisiamopronti, Nurse health professional, Professional nurse, Nurses by passion, NurseTimes, Nurse24.it, Nurse Specialist, Nurseallface, Nursing research, NursesInProgress, Nurses, Active Nurses, Nurses Italy, Nurses supporting health, Nursing Mobility, Nursing Competitions and Informed Nurses.

### 2.2. Participants

The questionnaire was addressed to all Italian nurses who were employed as nurses in both private and public healthcare services.

### 2.3. The Questionnaire

An ad hoc questionnaire was created, which included the first part concerning some sampling characteristics of the participants associated with their nursing activity, specifically including:Sex, if the respondent was female or male;Years of work experience, divided into two main subgroups, including less than 5 years and over 6 years;Shift work per day, whether the nursing work occurred in a single or double shift, whether during the morning and/or the afternoon (day), or during the night (night);Weight, expressed in kilograms (Kg), and height, expressed in meters (m), with the BMI value assessed as meters divided by the height squared. Each BMI value was classified according to the range groups explained in the current literature [53] as follows: for BMI values below 18.49, an underweight condition was classified; for values between 18.50 and 24.99, a normal weight condition was assessed; for BMI scores between 25 and 29.99, an overweight condition was identified; and for values over 30, an obesity condition was identified.

Then, the Depression, Anxiety and Stress Scale-21 Items (DASS-21) [54,55] were administered, including a total of 21 items which assessed the negative emotional conditions of depression, anxiety and stress, respectively. Each of the three DASS-21 subscales included 7 items, divided into subscales with similar content. The depression scale measured dysphoria, hopelessness, devaluation of life, self-deprecation, lack of interest/involvement, anhedonia and inertia. The anxiety scale assessed autonomic arousal, skeletal muscle effects, situational anxiety and the subjective experience of anxious affect. The stress scale measured levels of chronic non-specific arousal by assessing difficulty in relaxing, nervous arousal, and the states of being easily upset/agitated, irritable/over-reactive and impatient. Each item was associated with a 7-point Linkert scale, which ranged from “0”, indicating that the situation mentioned never happened, to ‘’3’’, indicating that the situation mentioned happened almost all the time. Finally, by summing items numbered 1, 6, 8, 11, 12, 14 and 18 and multiplying the total score by 2, the stress condition was assessed. By summing items numbered 2, 4, 7, 9, 15, 19 and 20 and multiplying the total score by 2, the anxiety condition was defined, and, finally, by adding items numbered 3, 5, 10, 13, 16, 17 and 21 and multiplying the total score by 2, the depression condition was defined. All three conditions, once defined, were classified into five levels, including normal, mild, moderate, severe and extremely severe.

### 2.4. Statistical Analysis

All data were developed through the Statistical Package for the Social Sciences (SPSS), version 20. All variables included in the present study were categorical variables and presented in frequencies and percentages. The chi-square test was performed for each psychological condition according to sex, work experience, shift and BMI levels. Finally, linear regressions were performed in order to explain any associations between anxiety, depression and stress conditions and sex, work experience, shift and BMI characteristics. All *p*-values of < 0.05 were considered as statistically significant.

### 2.5. Ethical Considerations

In the first part of the questionnaire, there was a clear explanatory note about the research study, detailing its relative purpose, as well as the possibility of joining and consenting to the processing of the data provided or not. All participants who did not give their consent were not enrolled in the study.

The present study was approved by the Ethical Committee of Polyclinic in Bari, Italy, with the protocol no. 0040/56/02/05/2022.

## 3. Results

In Italy, over 360,000 healthcare professionals were employed as nurses in all Italian healthcare facilities. The representative sample was assessed using 384 participants by considering a confidence level of 95% and a confidence interval of 5%. The sample was randomly collected until its completion. Therefore, the answer rate exceeded the representative sample, since it a total of 408 Italian nurses were recruited. Of these, 76% were females and 24% were males. 33.6% had been employed for less than 5 years and 66.4% had been working for over 6 years. 26.7% of the participants worked only day shifts and 73.30% of the interviewees worked also during night shifts.

Among the participants, with respect to their BMI scores, 4.4% were reported as being in an underweight condition, 57.6% had a normal weight condition, 26% had an overweight and 12% had an obese condition.

According to the anxiety levels registered, Table 1 shows all the sampling characteristics of the participants, highlighting significant differences in anxiety levels according to work experience, since nurses employed over 6 years reported higher anxiety levels than their younger colleagues (*p* = 0.035). No further significant differences between anxiety levels and the sampling characteristics were identified.

Considering the depression levels identified according to the sampling characteristics (Table 2), significant differences were registered according to sex (*p* = 0.017), as females reported having higher depression condition than males, ranging from moderate (14.50%) to severe (10.30%) and extremely severe (12.50%). Depression condition also significantly differed according to BMI levels (*p* = 0.003), as 5.90% of overweight and 2.50% of obese participants reported extremely severe depression conditions.

As regards the stress levels according to the sampling characteristics (Table 3), significant differences were registered according to BMI levels (*p* < 0.001), as overweight subjects reported having higher stress levels (7.40%) than the other subgroups.

Finally, considering the associations between anxiety, depression and stress conditions and sex, work experience, shift and BMI (Table 4), a significant relationship was identified between anxiety levels and shift work, as nurses who worked also during the night shift developed anxiety conditions more than their day shift-working colleagues (*p* = 0.023). No further associations were found.

## 4. Discussion

The present study aimed to assess any differences in anxiety, depression and stress conditions among nurses according to sex, years of work experience, shift and BMI sampling characteristics. Additionally, this research also evaluated any associations between the sampling traits and psychological conditions in order to establish the ultimate existence of a potential risk factor in the nursing profession, which could also contribute to worse mental health conditions. Specifically, the purpose of the present research was to investigate whether the nursing profession could itself be a potential risk factor for developing negative psychological conditions among nurses [56], exposing them to a great degree of psychosocial job-stress correlated with numerous health risks later in life [57,58]. In this way, chronic oxidative stress-induced cellular damage is involved in many debilitating diseases, such as heart disease and cancer [59,60].

In the present study, significant differences were registered between stress conditions according to BMI levels (*p* < 0.001), as overweight subjects reported higher stress levels (7.40%) than the other subgroups. The data were in agreement with the current literature, showing significant associations between high BMI values and psychosocial job-stress conditions. In a Canadian national population health survey, it was reported that men employed for more than 35 h/week appeared to be more overweight. In an American nurses’ health study [61,62], higher job pressure was also correlated with higher BMI levels. The age variable was also positively associated with stress in another similar study [63,64], with the data suggesting the increased prevalence of stress conditions among older nurses, influenced by age-related burn-out and a lower physical ability to perform at the highest level [65]. Evidence has also suggested a link between nursing job-stress and its related oxidative biomarkers (MDA, antioxidant capacity), as reported among 204 registered nurses from two Egyptian hospitals [15]. Another Chinese study suggested a significant association between occupational stress, oxidative stress and reduced antioxidant capacity in older nurses [66]. As well as stress, from the present data it emerged that there were higher levels of anxiety conditions among nurses who had been employed for more than 6 years compared to their younger colleagues. This data seemed to be in disagreement with the current literature, according to which younger nurses often experienced acute anxiety, stress or psychological disorders during the transition from being a student to an employed nurse, due to feelings of being unqualified, with scarce self-esteem in complex clinical situations, and the fact that they become more vulnerable to experiencing work-related stress, workplace violence, bullying or kinds of emotional abuse in interactions at work. Therefore, younger nurses became more isolated or stressed than their colleagues and peers [67].

In the present study, significant associations were identified between anxiety levels and shift conditions, since nurses who were employed also during the night shift reported higher stress levels than their colleagues who were employed only during the day shift. In this regard, too, the literature has suggested that shift work, working weekends and long working hours influence nurses’ health risks. In fact, by working during the night shift for many years of their lives, nurses might suffer from burn-out syndrome, accompanied by elevated oxidative stress biomarkers, as reported by several studies, from Spain [15] to Poland and Turkey [16,68]. Shift work was associated with reduced sleep duration and poor sleep quality, influencing physiological rhythms and desynchronizing the sleep cycle, hormonal rhythms and metabolic functions, such as daily cortisol rhythms, which were adjusted with alterations in the circadian sleep cycle [69], as reported also in salivary cortisol rhythm profiles, which differed among subjects working night shifts and those working day shifts [70,71]. Additionally, sleep restriction was associated with glucose intolerance, augmented cortisol and an increase in the sympathetic drive [72]. Therefore, highly important interactions were discovered between sleep deprivation, disturbed cortisol rhythm and metabolic complications, as reported in a meta-analysis of 34 observational studies involving over 2 million shift workers, which demonstrated a higher degree of myocardial infarctions and ischemic strokes [73], and particularly an increased risk of coronary events.

Recent studies demonstrated how an overweight condition and obesity were more prevalent in shift workers than in day workers, including in Mediterranean regions [41]. In addition, the literature reported that shift workers gained weight more frequently than daily workers [37,38,41,74,75], demonstrating a large variability in the prevalence of overweight conditions and obesity. In another study, this variability was influenced not only by shift work, but also by increasing age [76]. A study enrolling 787 day shift workers and 787 day–night shift workers highlighted that a longer exposure to shift work conditions associated with long working hours might predict a higher BMI value [77].

Therefore, the literature is in agreement in considering nursing shift work as a fundamental determinant of a nurse’s circadian dysregulation by identifying a greater number of physical and psychological symptoms with a higher risk of several chronic physiological and metabolic diseases [32,78], as a structure of the timing of rhythms was superimposed onto the human body with trends in development and ageing [28]. The temporal organization of organic activities estimated for each action was carried out during a particular hour of the day, by developing a series of controlled and rigorous procedures and determining the circadian rhythm expression proposed by Halberg [78].

According to the present data, depression conditions also significantly differed according to BMI levels, as 5.90% of overweight and 2.50% of obese participants reported extremely severe depression conditions.

In this regard, evidence has suggested that depressed people also exhibited an alteration in the HPA axis [79,80]. Specifically, mean cortisol levels during the whole day were elevated and similar in people with both unipolar and bipolar depression [81], deregulating the glucocorticoid and mineralocorticoid receptors in the brain, which may explain how the HPA axis is controlled and contribute to the clinical efficacy of antidepressant treatments [82].

The data highlighted differences in depression conditions according to sex (*p* = 0.017), as females reported higher depression conditions than males, ranging from moderate (14.50%) to severe (10.30%) and extremely severe (12.50%). Women seem to be at risk of becoming depressed [47], as reported in a recent Chinese cross-sectional study, which reported depression among female participants ranging from mild (61.7%) to moderate (25.1%) and severe (38.00%) [48]. Similar results were recorded in South Korea [83], in which approximately 38% of the enrolled nurses experienced depressive symptoms. The same trend was reported in the present study: females reported mild (7.80%) to moderate (4.50%) and severe (10.30%) or extremely severe (12.50%) depressive conditions, which were significantly higher than the levels reported by their male colleagues (mild: 10.05; moderate: 7.80%; severe: 2.70%; extremely severe: 4.90%).

Nevertheless, Wang et al. [84] argued that depressive symptoms were associated with work stress conditions as well.

### Limitations

The present study had some limitations. First of all, the sample was small. Moreover, its observational design did not allow for the complete assessment of the causal relationships between anxiety, depression and stress conditions and the selected sampling variables. Prospective research using more numerous samples collected in a greater number of healthcare environments will be helpful for investigating this topic in greater depth.

## 5. Conclusions

The data were in agreement with the current literature, indicating that nurses might take care not only of their patients but also of themselves, in both the physical and mental aspects [85]. Surely, such data will allow healthcare organizations to introduce preventive interventions for improving well-being among nurses by ameliorating their organizational working environments, and to consider individual nursing pathways that will also improve nurses’ psychological and physical well-being.

## Figures and Tables

**Table 1 diseases-10-00050-t001:** Sampling characteristics according to anxiety levels.

	Anxiety Levels					
Sampling Characteristics	Normal *n* (%)	Mild *n* (%)	Moderate *n* (%)	Severe *n* (%)	Extremely Severe *n* (%)	*p*-Value
**Sex**						
Female	93 (22.80)	27 (6.60)	59 (6.60)	42 (10.30)	89 (21.80)	0.571
Male	28 (6.90)	6 (1.50)	23 (5.60)	9 (2.20)	32 (7.80)	
**Work experience**						
>5 years	44 (10.80)	14 (3.40)	25 (6.10)	8 (2.00)	46 (11.30)	0.035 *
<6 years	77 (18.90)	19 (4.70)	57 (14.00)	43 (10.50)	75 (18.40)	
**Shift**						
Day	23 (5.60)	6 (1.50)	26 (6.40)	18 (4.40)	36 (8.80)	0.077
Night	98 (24.00)	27 (6.60)	56 (13.70)	33 (8.10)	85 (20.80)	
**BMI levels**						
Underweight	2 (0.50)	0 (0)	5 (1.20)	2 (0.50)	9 (2.20)	0.409
Normal weight	66 (16.20)	23 (5.60)	43 (10.50)	32 (7.80)	71 (17.40)	
Overweight	37 (9.10)	8 (2.00)	22 (5.40)	10 (2.50)	29 (7.10)	
Obese	16 (3.90)	2 (0.50)	12 (2.90)	7 (1.70)	12 (2.90)	

* *p* < 0.05 is statistically significant.

**Table 2 diseases-10-00050-t002:** Sampling characteristics according to depression levels.

	Depression Levels					
Sampling Characteristics	Normal *n* (%)	Mild *n* (%)	Moderate *n* (%)	Severe *n* (%)	Extremely Severe *n* (%)	*p*-Value
**Sex**						
Female	126 (30.90)	32 (7.80)	59 (6.60)	42 (10.30)	51 (12.50)	
Male	31 (7.60)	4 (1.00)	32 (7.80)	11 (2.70)	20 (4.90)	0.017 *
**Work experience**						
>5 years	44 (10.80)	14 (3.40)	25 (6.10)	8 (2.00)	21 (5.10)	0.032 *
<6 years	77 (18.90)	19 (4.70)	57 (14.00)	43 (10.50)	50 (12.30)	
**Shift**						
Day	23 (5.60)	6 (1.50)	26 (6.40)	18 (4.40)	22 (5.40)	0.873
Night	98 (24.00)	27 (6.60)	56 (13.70)	33 (8.10)	49 (12.00)	
**BMI levels**						
Underweight	5 (1.20)	0 (0)	5 (1.20)	6 (1.50)	2 (0.50)	0.003 *
Normal weight	96 (23.50)	19 (4.70)	51 (12.50)	34 (8.30)	35 (8.60)	
Overweight	36 (8.8)	17 (4.20)	24 (5.90)	5 (1.20)	24 (5.90)	
Obese	20 (4.90)	0 (0)	11 (2.70)	8 (2.00)	10 (2.50)	

* *p* < 0.05 is statistically significant.

**Table 3 diseases-10-00050-t003:** Sampling characteristics according to stress levels.

	Stress Levels					
Sampling Characteristics	Normal *n* (%)	Mild *n* (%)	Moderate *n* (%)	Severe *n* (%)	Extremely Severe *n* (%)	*p*-Value
**Sex**						
Female	76 (18.60)	54 (13.20)	79 (19.40)	66 (16.20)	35 (8.60)	0.141
Male	31 (7.60)	14 (3.40)	18 (4.40)	17 (4.70)	18 (4.40)	
**Work** **experience**						
>5 years	44 (10.80)	14 (3.40)	25 (6.10)	8 (2.00)	14 (3.40)	0.812
<6 years	77 (18.90)	19 (4.70)	57 (14.00)	43 (10.50)	39 (9.60)	
**Shift**						
Day	27 (6.60)	24 (5.90)	19 (4.70)	23 (5.60)	13 (3.90)	0.234
Night	80 (19.60)	44 (10.80)	78 (19.100)	60 (14.70)	37 (9.10)	
**BMI levels**						
Underweight	0 (0.50)	0 (0)	9 (2.20)	7 (1.70)	0 (0)	>0.001 *
Normal weight	57 (14.00)	46 (11.30)	64 (15.70)	49 (12.00)	19 (4.70)	
Overweight	34 (8.30)	17 (4.20)	6 (1.50)	19 (4.70)	30 (7.40)	
Obese	14 (3.40)	5 (1.20)	18 (4.40)	8 (2.00)	4 (1.00)	

* *p* < 0.05 is statistically significant.

**Table 4 diseases-10-00050-t004:** Linear regression between the sampling characteristics and anxiety, depression and stress conditions.

Psychological Conditions	Sex	Work Experience	Shift	BMI
**Anxiety**				
β	0.025	0.024	−0.113	−0.087
t	0.503	0.484	−2.280	−1.772
p-value	0.616	0.628	0.023 *	0.077
**Depression**				
β	0.089	−0.082	−0.019	0.028
t	1.790	−1.643	−0.382	0.560
p-value	0.074	0.101	0.703	0.576
**Stress**				
β	−0.004	0.033	−0.066	−0.003
t	−0.084	0.665	−0.118	−0.057
p-value	0.933	0.566	0.906	0.955

* *p* < 0.05 is statistically significant.

## Data Availability

Data are available from the corresponding author.

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
