# Peer review of "Work Conditions of Italian Nurses and Their Related Risk Factors: A Cohort Investigatory Study"

_diseases, 2022, doi:10.3390/diseases10030050_

Round 1

Reviewer 1 Report

Thank you for the opportunity to review this paper which aimed to investigate differences in anxiety, depression and stress conditions in nurses according to sex, work experience, shift and BMI. I really appreciated the commitment of the author, who conducted and wrote the work by herself, and you can fell that this is a subject that she cares much about. Nonetheless, I have numerous comments to make on this paper.

The Introduction is informative but a bit too long and redundant: from line 39 to 68 the author should try to synthetize the concepts and re-organize them in a more concise way - as a reader there is the feeling that some information are provided more than once and a lot of attention is given to the role of shifts. The Aim section could be included in the Introduction.

Materials and Methods are clearly described, my only suggestion is to provide with more information on the way the questionnaire was distributed and participants recruited ('nursing social and pages' is a bit too vague). Why did the author decide not to include age as an information to be collected?

As regards Results: it is not correct to start a sentence with a numeral. The first or second decimal of a digit should be deleted if it is a zero (e.g. 76.60 should be 76.6, 24.00 should be 24, etc.).

The are several typos to be corrected in the first section of the Discussion (line 179-191). I have the feeling the author pasted some sentences without double checking before final submission.

Section 6 has been kept by mistake.

Extensive editing of English language is required: the author must have the manuscript edited from a professional as language is very poor.

There are typos to be corrected in the abbreviations over the text (e.g. HPA, BMI - before using the acronym, it should be quoted in full).

This work can be re-considered only after a major revision.

Author Response

xcvx

Rebuttal letter

“Work conditions in Italian nurses and its relating risk factors: a cohort investigatory study”

We thank the Referee for the valuables comments.

Reviewer 1

Thank you for the opportunity to review this paper which aimed to investigate differences in anxiety, depression and stress conditions in nurses according to sex, work experience, shift and BMI. I really appreciated the commitment of the author, who conducted and wrote the work by herself, and you can fell that this is a subject that she cares much about. Nonetheless, I have numerous comments to make on this paper.

R1: The Introduction is informative but a bit too long and redundant: from line 39 to 68 the author should try to synthetize the concepts and re-organize them in a more concise way - as a reader there is the feeling that some information are provided more than once and a lot of attention is given to the role of shifts. The Aim section could be included in the Introduction.

A1: The Introduction section was performed according to the Reviewer’s suggestion.

R2: Materials and Methods are clearly described, my only suggestion is to provide with more information on the way the questionnaire was distributed and participants recruited ('nursing social and pages' is a bit too vague). Why did the author decide not to include age as an information to be collected?

A2: The Author included years of work experience as directly associated to the nursing profession, independently from their really age of all participants.

R3: As regards Results: it is not correct to start a sentence with a numeral. The first or second decimal of a digit should be deleted if it is a zero (e.g. 76.60 should be 76.6, 24.00 should be 24, etc.).

A3: All the Results section was implemented according to suggestions.

R4: The are several typos to be corrected in the first section of the Discussion (line 179-191). I have the feeling the author pasted some sentences without double checking before final submission.

Section 6 has been kept by mistake.

Extensive editing of English language is required: the author must have the manuscript edited from a professional as language is very poor.

There are typos to be corrected in the abbreviations over the text (e.g. HPA, BMI - before using the acronym, it should be quoted in full).

A4: All the manuscript was revised in order to delete all mistakes included in the first version.

Changes and modifications have been marked in red color in the text to speed up reviewing.

I hope that the revised version of our manuscript, now, is ready for acceptance.

Sincerely yours,

Elsa Vitale

Bari, 29.07.2022

Reviewer 2 Report

Nursing is a lifesaving and highly satisfying profession, yet it is considered one of the most stressful occupations. This study aimed to assess differences in anxiety, depression and stress states among nurses according to gender, work history, shift and BMI characteristics and their associations.

Title

Rather than ”psychological alterations”, the study can be perceived as a study of the impact of the work environment, or simply a survey of nurses' stress levels and a search for their relevant factors. Please reconsider the title.

Abstract

l  Abbreviations should be spelt out on first appearance (BMI).

l  Consider including the number of survey responses and response rates in the results section.

l  Describe how stress levels are assessed.

Introduction

It is unclear what has not been clarified in previous reports and what the novelty of this study is.

Methods

Well described

Results

l  Align the significant digits (number of digits).

l  Please specify whether all respondents completed the questionnaire, the response rate and the number of accesses.

l  With regard to shifts, it is curious that all participants are clearly segregated into 'Daily' and 'Night'. Do they work exclusively as 'Daily' or 'Night'?

l  Table 4Has multicollinearity been checked?

Author Response

xc

Rebuttal letter

Work conditions in Italian nurses and its relating risk factors: a cohort investigatory study.

We thank the Referee for the valuables comments.

Reviewer 2

R1: Nursing is a lifesaving and highly satisfying profession, yet it is considered one of the most stressful occupations. This study aimed to assess differences in anxiety, depression and stress states among nurses according to gender, work history, shift and BMI characteristics and their associations.

 A1: This sentence was paste in the abstract section.

R2: Title

Rather than “psychological alterations”, the study can be perceived as a study of the impact of the work environment, or simply a survey of nurses' stress levels and a search for their relevant factors. Please reconsider the title.

A2: The title was revised according to the reviewer’s suggestion.

R3: Abstract

l  Abbreviations should be spelt out on first appearance (BMI).

l  Consider including the number of survey responses and response rates in the results section.

l  Describe how stress levels are assessed.

A3: Abstract was revised.

R4: Introduction

It is unclear what has not been clarified in previous reports and what the novelty of this study is.

 A4: An explanation about the novelty of the study was stated in the Introduction section.

R5: Methods

Well described.

R6: Results

l  Align the significant digits (number of digits).

l  Please specify whether all respondents completed the questionnaire, the response rate and the number of accesses.

l  With regard to shifts, it is curious that all participants are clearly segregated into 'Daily' and 'Night'. Do they work exclusively as 'Daily' or 'Night'?

l  (Table 4)Has multicollinearity been checked?

 A6: significant digits are aligned. All information on sample size were included. All participants interviewed worked or during the morning and/or the afternoon and during the night shift.

In linear regressions performed, multicollinearity was checked (VIF<5), so the Author decided to not include it in the table:

VIF-sex: 1.003

VIF-work experience: 1.006

VIF-shift: 1.003

VIF-BMI: 1.003

Changes and modifications have been marked in red color in the text to speed up reviewing.

I hope that the revised version of our manuscript, now, is ready for acceptance.

Sincerely yours,

Elsa Vitale

Bari, 29.07.2022

Round 2

Reviewer 1 Report

The author has improved the manuscript and complied with the reviewers' suggestions. I would ask to have English checked by a professional (I doubt the author had the time to actually do it in such a short time frame), language does not sound proper in several parts..